# Scaling Temporal and Volumetric Datasets for Tumor Localization with Weak Annotations

**Yu-Cheng Chou** [1]                                    JOHNSON111788@GMAIL.COM
**Bowen Li** [1]                                            LBWDRUID@GMAIL.COM
**Deng-Ping Fan** [2]                                        DENGPFAN@GMAIL.COM
**Alan Yuille** [1]                                          AYUILLE1@JHU.EDU
**Zongwei Zhou** [1,*]                                       ZZHOU82@JH.EDU

[1] *Department of Computer Science, Johns Hopkins University, Baltimore, USA*

[2] *Computer Vision Lab, ETH Zürich, Zürich, Switzerland*

**Editors:** Under Review for MIDL 2024

## Abstract

Creating large-scale, well-annotated datasets is vital for training AI algorithms in tumor detection. However, with limited resources, it is challenging to determine the best type of annotations when annotating massive amounts of unlabeled data. To address this issue, we focus on polyps in colonoscopy videos and pancreatic tumors in abdominal CT scans, both requiring extensive pixel-wise annotation due to the high dimensional nature of the data. In this paper, we develop a new annotation strategy, termed Drag&Drop, which simplifies the annotation process to drag and drop, proving more efficient for temporal and volumetric imaging. Furthermore, we introduce a novel weakly supervised learning method based on the watershed algorithm to leverage Drag&Drop annotations. Experimental results show that, with limited resources, allocating weak annotations from diverse patients enhances model robustness more effectively than per-pixel ones on a limited set of images. In summary, this research proposes an efficient annotation strategy that is useful for creating large-scale datasets for screening tumors in various medical modalities.

**Keywords:** Weak annotation, detection, segmentation, colonoscopy, abdomen.

## 1. Introduction

Tumor detection and localization are often approached as a semantic segmentation task known as *detection by segmentation*. The hypothesis is that identifying and delineating tumor boundaries can improve the tumor detection rate (Xia et al., 2022). However, this idea might not apply to all medical scenarios, particularly for screening purposes, in which it is more critical to predict the approximate location and size of the tumors rather than focusing on the accurate segmentation of tumor boundaries. For instance, in polyp detection, precise boundary segmentation may not be crucial since the polyp can be removed during colonoscopy (Winawer et al., 1993; Rex et al., 2017). Yet, many public datasets provide per-pixel annotations for each polyp (Jha et al., 2020; Vázquez et al., 2017), which is time-consuming and costly, potentially wasting resources for large-scale tumor detection datasets. We posit that for certain detection tasks, high precision in boundary segmentation is not crucial, and therefore per-pixel annotations may not be necessary (Tables 1).

---

[*] Corresponding author

In this paper, we design a new weak annotation strategy for high-dimensional data by exploiting contextual information across dimensions. We call this strategy "**Drag&Drop**" because it involves clicking on the tumor and then dragging and dropping to provide the approximate radius of the tumor. To utilize **Drag&Drop** annotations, we further develop a weakly supervised framework based on the classical watershed algorithm, and it is optimized using the approximate tumor size and location constraints provided by **Drag&Drop**. We demonstrate in the experiments that training using the weak annotations by **Drag&Drop**, AI algorithms can perform similarly to pixel-wise annotations in tumor localization tasks.

## 2. Method

Unlike other strategies (Chu et al., 2021; Zhang et al., 2020; Cheng et al., 2022; Li et al., 2022) requiring multiple 2D annotations, the proposed **Drag&Drop** requires just one 2D label per lesion in the high-dimensional data, greatly reducing labeling time. To ensure quality, annotators must follow these steps: first, locate lesions and estimate their size and center by screening through multiple scans. Second, annotate lesions conveniently in arbitrary dimensions. Lastly, refine annotations by adding and removing annotation masks with a simple click on the foreground and background, respectively.

Next, we adopt a marker-based watershed transformation algorithm (Meyer, 1994) to propagate the **Drag&Drop** annotations to pseudo labels. Compared to other segmentation methods, the watershed algorithm does not require parameter tuning and can use markers as a form of user guide to refine the segmentation boundaries. The watershed algorithm views an image as a topographic landscape with ridges and valleys. In specific, given an input $I$, lesion markers $m_i^l$, and background markers $m_j^b$, the watershed line can be defined as the set of points of the support of $I$ that do not belong to any catchment basin $CB()$:

$$Wsh(I, m_i^l, m_j^b) = support(I) \cap \left[ \bigcup_i \left( CB\left(m_i^l\right)\right) \bigcup_j \left( CB\left(m_j^b\right)\right) \right]. \tag{1}$$

To segment the boundary via watershed segmentation, we dilate the **Drag&Drop** annotations into lesion and background markers in the 3D space as the initial flooding points. Given a central point and radius of a lesion, we first generate a 3D sphere and sample set the surface of the sphere as the background markers $m_j^b$. Specifically, we set $j$ as the number of integer points on the surface and randomly sample the background markers. Then we adaptively dilate the central point according to a certain ratio $N$ of the given radius to avoid under-segmentation and create the lesion marker $m_i^l$, ensuring that the result adequately covers the region of interest. To remove noise while preserving the boundary information, we further applied a morphological gradient to the input $I$ before watershed segmentation.

Even though the watershed algorithm can detect the boundaries of lesions, it remains vulnerable to noise in the image. To this end, we propose a masked back-propagation that computes the gradient only on the lesion and background regions; no gradient on the uncertain regions. In detail, we dilate pseudo labels with $M$ kernel size and ignore the gradient of the dilated area (which is the uncertain region) during back-propagation. In addition, due to the absence of per-pixel labels, the performance validation of the model during training leads to the misestimation of the model's performance. To obtain optimal detection performance, we propose to adopt a lesion-wise metric for validation purposes.

Table 1: Given a certain annotation budget, our **Drag&Drop** strategy outperforms the per-pixel annotation in tumor detection. More importantly, **Drag&Drop** improves pancreatic tumor segmentation from 0.43 to 0.54 measured by DSC scores.

| Method | Strategy | Lesion-level | | | | Patient-level | | | |
|--------|----------|-------------|-------------|-----------|----------|-------------|-------------|-----------|----------|
| | | Sensitivity | Specificity | Precision | F1-score | Sensitivity | Specificity | Precision | F1-score |
| **JHH** (Xia et al., 2022) | | | | | | | | | |
| nnUNet | Per-pixel | 0.611 | 0.339 | 0.422 | 0.522 | 0.765 | **0.702** | 0.752 | 0.613 |
| **nnUNet** | **Drag&Drop** | **0.715** | **0.429** | **0.575** | **0.610** | **0.886** | 0.649 | **0.749** | **0.735** |
| **SUN-SEG** (Ji et al., 2022) | | | | | | | | | |
| PNS+ | Per-pixel | 0.681 | 0.589 | 0.434 | 0.546 | 0.759 | 0.698 | 0.543 | 0.621 |
| **PNS+** | **Drag&Drop** | **0.719** | **0.677** | **0.512** | **0.595** | **0.804** | **0.790** | **0.644** | **0.668** |

## 3. Experiments & Results

***Benchmarking.*** We adopt two large-scale high-dimensional datasets in our experiments, including a private 3D volumetric CT dataset (*i.e.*, JHH (Xia et al., 2022)) and a colonoscopy video dataset (*i.e.*, SUN-SEG (Ji et al., 2022)). JHH dataset includes $2,426$ CTs of $1,213$ patients with per-pixel annotation of pancreatic tumor, containing classes of pancreas, PDAC, and Cyst. We randomly split 1683, 420, and 323 cases for training, validation, and testing, respectively. SUN-SEG dataset, on the other hand, offers $1,106$ video clips with $158,690$ frames total. We follow the same data splitting protocol in (Ji et al., 2022) and re-split the videos in a finer granularity, resulting in 507, 126, and $1,050$ cases respectively for training, validation, and testing. For pancreatic tumor and polyp detection, we adopt the state-of-the-art per-pixel segmentation methods (*i.e.*, nnUNet (Isensee et al., 2021) and PNS+ (Ji et al., 2022), respectively). We follow the default training setting and set $N = 0.2$ and $M = 0.5$ after extensive ablation studies.

***Population diversity.*** To evaluate the performance gain provided by data diversity in our **Drag&Drop** strategy, we calculate the number of per-pixel labels that can be annotated in the same amount of time. The annotation time can be defined as detection time plus the manual labeling time. Since pancreatic tumors are more camouflaged and require more time to be detected, we set the detection time as $30''$ and $2''$ for pancreatic tumors and polyps, respectively. With the manual labeling time of $4'15''$ and $2''$, we can have the number of per-pixel labeled and **Drag&Drop** labeled cases for pancreatic tumor (131 vs. $1,168$) and polyp detection (33 vs. 189). The results in Table 1 illustrate that the utilization of weak annotations can promote greater diversity and thus enhance results.

## 4. Conclusion

We propose **Drag&Drop**, a novel annotation strategy, and a weakly supervised framework for simplifying the annotation of medical images. Our proposed strategy can reduce 87.5% and 99.2% annotation efforts for polyp and pancreatic tumor detection, respectively, compared with per-pixel annotations. The experimental results demonstrate that using weak annotations from a larger data population, given a certain annotation budget, improves the model generalizability compared to per-pixel annotations from a small dataset.

## Acknowledgments

This work was supported by the Lustgarten Foundation for Pancreatic Cancer Research and the Patrick J. McGovern Foundation Award.

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
