# OpenReview forum: "Scaling Temporal and Volumetric Datasets for Tumor Localization with Weak Annotations"
_MIDL.io/2024/Short_Papers — MIDL 2024 Short Papers_

### Official Review · Reviewer_phKm · 2024-04-23

**Confidence:** 5
**Final Rating:** 3.5

**Review:**

This paper proposed a Drag&Drop annotation method with a watershed algorithm for the weakly supervised learning. Subsequently, the author evaluated the proposed method in two large datasets: JHH and SUN-SEG. However, due to missing details in the experiment, it is challenging for me to interpret the results of the experiments. First, what is the total annotation budget? What is the annotation time for the Drag&Drop method? Do you have any references for your estimation of “the manual labeling time of 4′15′′ and 2′′ ”? Maybe use the minutes and seconds rather than the ′ and ′′ symbols to avoid confusion.  More importantly, what are the detection/localization success criteria for the proposed method and comparison methods?  Without sufficient details, interpreting the results presented in Table 1 is somewhat challenging. In addition, I would like to suggest including some visualization examples and providing more discussion of Table 1.

---

### Decision · Program_Chairs · 2024-04-26

Accept